# Current characteristics of animal rabies cases in Thailand and relevant risk factors identified by a spatial modeling approach

**Weerapong Thanapongtharm[1], Sarin Suwanpakdee[2], Arun Chumkaeo[3], Marius Gilbert[4,5], Anuwat Wiratsudakul[2]***

**1** Department of Livestock Development (DLD), Bangkok, Thailand, **2** Department of Clinical Sciences and Public Health, and the Monitoring and Surveillance Center for Zoonotic Diseases in Wildlife and Exotic Animals, Faculty of Veterinary Science, Mahidol University, Nakhon Pathom, Thailand, **3** Songkhla Provincial Livestock Office, Songkhla, Thailand, **4** Spatial Epidemiology Lab. (SpELL), University of Brussels, Brussels, Belgium, **5** Fonds National de la Recherche Scientifique (FNRS), University of Brussels, Brussels, Belgium

* anuwat.wir@mahidol.edu

**Data Availability Statement:** Some restrictions will apply as some data are sensitive, for example, distances to the country's borders. The data

## Abstract

The situation of human rabies in Thailand has gradually declined over the past four decades. However, the number of animal rabies cases has slightly increased in the last ten years. This study thus aimed to describe the characteristics of animal rabies between 2017 and 2018 in Thailand in which the prevalence was fairly high and to quantify the association between monthly rabies occurrences and explainable variables using the generalized additive models (GAMs) to predict the spatial risk areas for rabies spread. Our results indicate that the majority of animals affected by rabies in Thailand are dogs. Most of the affected dogs were owned, free or semi-free roaming, and unvaccinated. Clusters of rabies were highly distributed in the northeast, followed by the central and the south of the country. Temporally, the number of cases gradually increased after June and reached a peak in January. Based on our spatial models, human and cattle population density as well as the spatio-temporal history of rabies occurrences, and the distances from the cases to the secondary roads and country borders are identified as the risk factors. Our predictive maps are applicable for strengthening the surveillance system in high-risk areas. Nevertheless, the identified risk factors should be rigorously considered and integrated into the strategic plans for the prevention and control of animal rabies in Thailand.

## Author summary

Rabies is a deadly viral zoonotic disease responsible for thousands of deaths worldwide. The disease is highly prevalent in developing countries including Thailand. The main reservoir hosts in these settings are dogs. Currently, there is a global effort to stop human deaths from dog-mediated rabies by 2030. To achieve this goal, scientifically driven policies must be rigorously implemented. In this study, we used data on rabies outbreaks in the high-prevalent years in Thailand together with other related factors to describe important characteristics of the outbreaks and to identify the main factors that contribute to the

underlying the results presented in the study are available from The Department of Livestock Development, Thailand (foreign@dld.go.th). An official letter is required for requesting the data.

**Funding:** AW received a financial support from the National Science and Technology Development Agency (NSTDA), Thailand (Grant ID. P-18-51758). The funders had no role in study design, data collection and analysis, decision to publish, or preparation of the manuscript.

**Competing interests:** The authors have declared that no competing interests exist.

higher risk of the outbreaks across space and time. Surprisingly, we found that owned dogs were more affected than the stray ones and the peak of the outbreaks was identified in winter rather than summer. There are many spatial factors involved in the spread of animal rabies, for example, history of rabies epidemics in the areas. The current policy planning and implementation should be revised based on scientific evidence to prevent such repeated outbreaks.

## Introduction

Rabies is a zoonotic disease caused by the Rabies virus belonging to the genus *Lyssavirus* of the family *Rhabdovirudae* [1]. World health organization (WHO) estimated that it caused almost sixty thousand deaths in humans per year in over 150 countries worldwide [2]. The majority of human cases were caused by dog-mediated rabies and mostly occurred in rural poor populations particularly in Asia and Africa, while minority cases were bat-mediated occurring in the Americas [2]. Many countries have been declared free from dog-mediated rabies such as the United States, Canada, Japan, Australia, and some Latin American countries. However, the transmission in wildlife and imported cases have still been reported in many of those countries [2].

A trend of human rabies in Thailand has gradually declined over the past four decades while a tendency of animal rabies has increased for the last ten years. Rabies was included in the list of notifiable diseases in Thailand in 1980 with 370 human cases reported in that year. Subsequently, 185, 50, 15, and 2 human rabies cases were reported in 1990, 2000, 2010, and 2020, respectively [3,4]. A tendency of animal rabies had a coincidence with human rabies until 2013. There were 2,939 animal cases reported in 1995, reducing to 1,181 cases in 2000, followed by 249 and 117 cases in 2010 and 2013, respectively [3,4]. However, the number of animal cases reach 250 in 2014. It then gradually increases to 1,105 cases in 2018 [5].

The successful prevention of human rabies depends on the effective control of the disease in domestic dog populations [6,7]. Dogs are considered the principal reservoirs responsible for the overwhelmed reported animal and human rabies cases [6,8–10]. A rabies control program should take both technical and socio-cultural frameworks into account [7]. The former includes vaccination programs, appropriate diagnostic capacity, disease surveillance programs, dog population management, and animal movement control. The latter is composed of public awareness, promoting responsible pet ownership and animal welfare. The strategies to control rabies in Thailand have been continuously implemented under laws and regulations by collaboration among governmental agencies, private sectors, and the general public [11]. Eight strategies under the present strategic plan of rabies control include i) the surveillance, prevention, and control of rabies in animals, ii) the management of animal shelters, iii) the surveillance, prevention, and health care of rabies in humans, iv) the driving of implementation of rabies in local areas, v) the public relations, vi) the integration and management of rabies information, vii) the monitoring and evaluation, viii) the innovation and technology transfer [12].

The surveillance of animal rabies plays an important role in disease detection, resulting in effective prevention and control in both animal and human rabies. Risk-based approaches have been used to enhance veterinary surveillance by identifying surveillance needs and priorities. These approaches are helpful for effective resource allocation and increase the chances of disease detection [13,14]. Spatial risk maps have been applied in several epidemiological surveillance, for example, highly pathogenic avian influenza (HPAI) [15–18], porcine reproductive and respiratory syndrome (PRRS) [19], Nipah virus infection [14,20]. This study,

therefore, aimed at providing a spatial risk-based assessment to improve the surveillance, prevention, and control of animal rabies which is one of the eight strategies implemented by the Thai government [12]. Our objectives were divided into two folds namely; i) to describe the characteristics of animal rabies cases in Thailand in 2017–2018 focusing on the animals affected by rabies and their temporal and spatial distributions, and ii) to identify the spatial and temporal risk factors influencing the distribution of rabies in Thailand. These allowed us to observe the distribution patterns, the important factors relevant to animal rabies occurring in high prevalence years, and to predict the occurrence of rabies in advance.

## Methods

### Descriptive analysis

The data on animal rabies cases during 2017–2018 were used in this study to describe the important characteristics and the spatio-temporal distribution patterns of the outbreaks. These data were derived from two surveillance programs namely passive and active surveillance, in which the samples were submitted to nine accredited laboratories under the Department of Livestock Development (DLD) and another laboratory belonging to the Queen Saovabha Memorial Institute. In the passive surveillance, the owners of animals or veterinarians submitted animal samples (carcasses or heads) to laboratories. Most of these cases were suspicious of rabies infection. In contrast, the DLD officers, in the active surveillance, were compulsory to collect samples of animal carcasses or heads that died with inconclusive symptoms such as by car accident or in veterinary clinics or hospitals. Those samples were examined with the Fluorescent antibody test (FA test) and the results were reported via a web-based reporting system called "ThaiRabies.net" [5]. We then extracted these data and used them in our analysis. We only used secondary data supplied through the system. No human or animal subjects were involved. The study thus required no human or animal ethical approvals. However, the study designs and protocols used were approved by the Department of Livestock Development, Thailand.

### Cluster analysis

The presence of spatial clusters of rabies cases in 2017 and 2018 in Thailand was analyzed based on the spatial scan statistic proposed by Kulldorff and Nagarwalla [21], using the centroids of sub-districts as the locations of the observations. The cases were divided into two periods (2017 and 2018) to differentiate clusters occurring within a particular epidemic period. The SaTScan version 9.3 software was implemented with the following settings for purely spatial, Bernoulli model, to scan for the area with high rates of infection with 999 replications of Monte Carlo simulations. The maximum percentage of the population at risk included in a cluster was 1%.

### Spatio-temporal modeling

We used a spatial distribution model called "Generalized Additive Model (GAM)" to quantify the association between rabies occurrences and predictive factors in the sub-district level (Thai administrative units contain 4 levels composed of 77 provinces, 926 districts, 7,416 sub-districts, and 74,944 villages). The predictive factors were separated by months, in which the factors in each month were used to predict the rabies occurrence in the following month. For example, we used the predictive factors of February 2017 to predict the rabies occurrence in March 2017 and so on. GAM, an extension of the Generalized Linear Models (GLM), allowed us to access the non-linear relationship between the response variable and multiple

explanatory variables through smoothing functions [22,23]. GAM was employed in this study as it was more flexible than GLM and the complex functions could be fitted as much as performed with machine learning methods [22]. Besides, different risk factors could be simultaneously identified with the model as required by epidemiologists. The formula of the GAM is as shown below.

$$log(\mu_i) = \beta_0 + f_1(X_{1i}) + \ldots + f_m(X_{mi}) + \varepsilon_i$$

, where $\mu_i = \frac{P_i}{1-P_i}$, and $P_i$ is the probability of the outcome (rabies occurrence), $\beta_0$ is a constant (called intercept) indicating the value of response variable when $X = 0$, $X = (X_1, \ldots, X_m)$ is a vector of m predictor variables, and $f = (f_1, \ldots, f_m)$ is a vector of $m$ smoothing curve, and $\varepsilon$ is the error or unexplained information.

Our data was then separated into two sets including the model set (data between January 2017 and January 2018) and the test set (data between January 2018 to January 2019). The number of animal rabies cases was defined as a dependent variable while the set of predictive factors (independent variables) included i) demographic data (human population, dog populations (owned dogs and ownerless dogs), and cattle population), ii) geographic data (length of 4 types of the road; the main road, the secondary road, the concession road, and the local road), iii) distance from the case locations to the country border, and iv) history of rabies occurrences (in months, rabies occurring in their sub-district within a month, and rabies occurring in at least a neighboring sub-district in the same period). A human population density raster map at 100-m resolution was obtained from the Worldpop project (https://www.worldpop.org/). Dog population data were obtained from ThaiRabies.net, which have been annually surveyed and reported via the system by Local Administration Organizations (LAOs). Geographic data including administrative units and roads were provided by the Land Development Department (LDD) [24]. Data on the monthly situation of rabies that occurred in neighboring sub-districts in the past month were processed using; i) igraph package [25] in R to build connectivity matrix between each sub-district and its neighbors, ii) a data frame combining all sub-districts, and iii) binary identification (0 and 1); if there was at least a neighboring sub-district was found with rabies case(s) in the month earlier, those sub-districts were coded as 1 (and 0, otherwise).

We processed our model in two steps; i) selecting significant variables and ii) bootstrapping the analyses. To select the significant variables, a univariate logistic regression was used to screen all variables using GAM, keeping only those associated with the outcome of a *p-value* ≤ 0.1. All variables with a significant p-value of 0.1 were chosen in the GAM, and subsequently removed one by one, starting from the variable with the lowest contribution to the model. This procedure continued until all variables were significant at the p-value of 0.05. A previous study suggested that logistic regression tended to be biased when the prevalence in the dependent variable was lower than 10% [26]. Nine times the number of positive cases were hence randomly selected at each bootstrap to maintain 10% of the positive values of the outcome variable. This approach was conducted in both model and test sets. To prevent over-fitting, GAM was subject to bootstrapping of the analyses over 100 repetitions [27].

To evaluate the predictive power of the models, the areas under the curve (AUC) of the receiver operating characteristic plots (ROC) were generated. The AUC is a quantitative measure of the overall fit of the model that varies from 0.5 (chance event) to 1.0 (perfect fit) [28]. In our study, AUC values of 0.5–0.7 indicated a low accuracy, values of 0.7–0.9 indicated high accuracy, and > 0.9 pointed out a very high accuracy [29,30]. Our evaluation process was carried out twice in i) the whole studied period of the model and test sets and ii) each month of

both sets. In the end, we also swapped between the model set and the train set to see another way around.

## Results

### Descriptive analysis

In total, 8,574 and 9,601 animal samples were submitted for rabies examination in 2017 and 2018, respectively. Of these, 848 (9.9%) and 1,475 (15.4%) turned positive. In both years, dogs were identified as the main affected species, followed by cattle, cats, and others. However, the number of infected cattle in 2018 was slightly increased (Table 1). Surprisingly, we found that over half of the rabies-positive animals were owned, and unvaccinated animals (particularly dogs) were mostly infected. The history of animal-keeping practices revealed that semi-free roaming (occasionally confined) was the main infected group, followed by free-roaming and confined animals, respectively.

The temporal and spatial distributions of animal rabies cases in Thailand in 2017 and 2018 are depicted in Fig 1. In terms of temporal distribution, the number of positive cases was relatively high at the beginning of both years and gradually decreased until June. Subsequently, the number increased again in July and August. Nevertheless, we observed that between September and December in both years, the patterns of rabies positivity were different. In 2017, the number of animal rabies cases was sharply increased until December whereas it was gradually decreased in 2018.

In the spatial distribution, animal rabies cases in 2017 (Fig 1B) were most prevalent in the northeast followed by the central and the south of Thailand. In 2018, the rabies cases still

**Table 1. Descriptive characteristics of animals infected with rabies virus in Thailand, 2017–2018.**

| Characteristic | Year | |
| --- | --- | --- |
| | **2017** | **2018** |
| **Type of animals** | | |
| Dog | 742 (87.5%) | 1,286 (87.2%) |
| Cattle | 57 (6.7%) | 118 (8%) |
| Cat | 41 (4.8%) | 51 (3.5%) |
| Others | 8 (0.9%) | 20 (1.4%) |
| **Ownership** | | |
| Owned animals | 449 (52.9%) | 839 (56.9%) |
| Stray animals | 297 (35.0%) | 502 (34.0%) |
| Unidentified | 102 (12.0%) | 134 (9.1%) |
| **Vaccination against rabies** | | |
| Unvaccinated | 337 (39.7%) | 575 (39.0%) |
| Vaccinated | 84 (9.9%) | 171 (11.6%) |
| Vaccinated < 1 month | 14 (1.7%) | 47 (3.2%) |
| Vaccinated > 1–6 months | 14 (1.7%) | 55 (3.7%) |
| Vaccinated > 6–12 months | 25 (2.9%) | 26 (1.8%) |
| Vaccinated > 12 months | 31 (3.7%) | 43 (2.9%) |
| Unidentified | 427 (50.4%) | 729 (49.4%) |
| **Animal-keeping practices** | | |
| Confined animals | 39 (4.6%) | 67 (4.5%) |
| Semi-free roaming animals | 153 (18.0%) | 299 (20.3%) |
| Free-roaming animals | 70 (8.3%) | 120 (8.1%) |
| Unidentified | 586 (69.1%) | 989 (67.1%) |

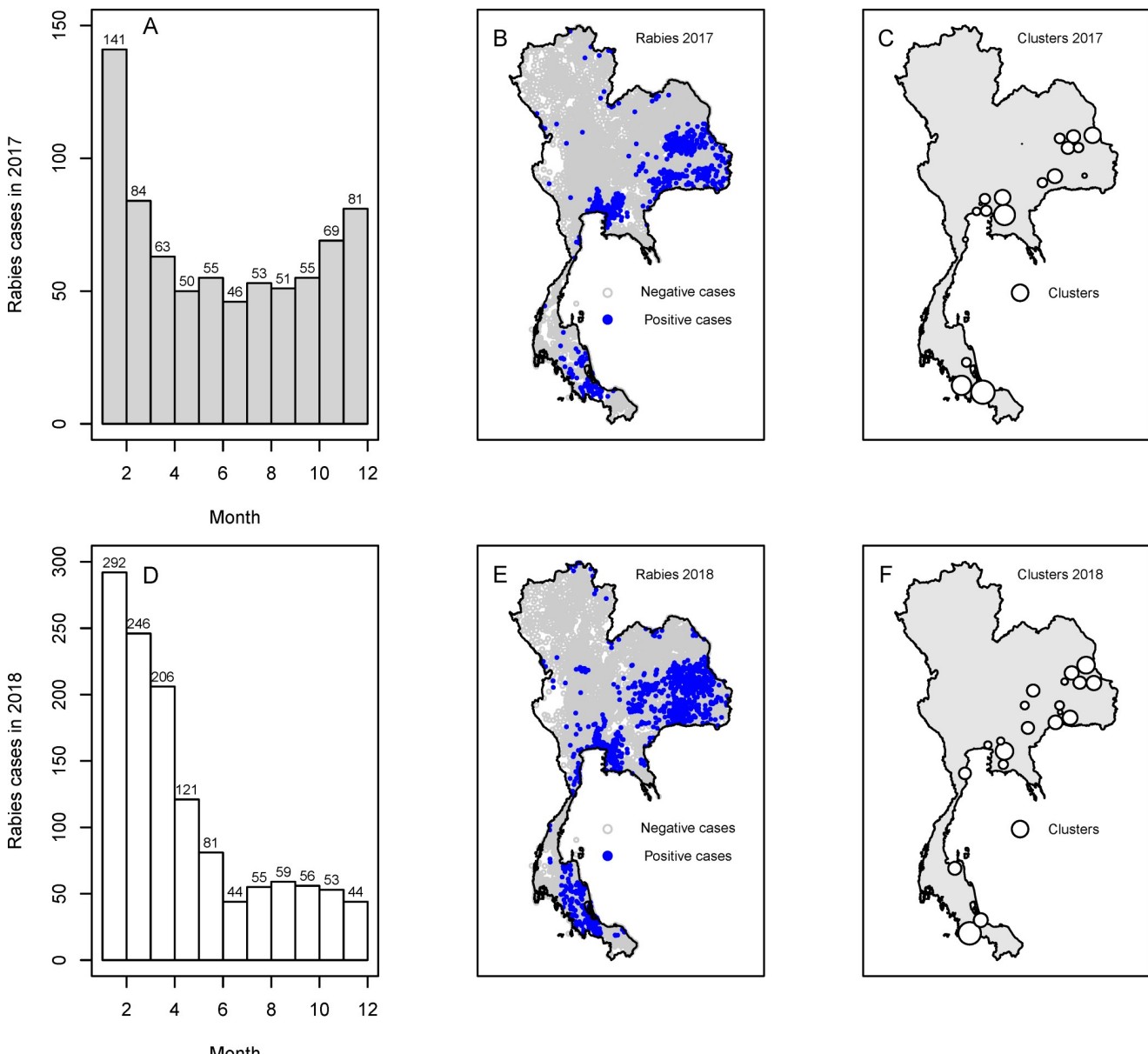

**Fig 1. Spatio-temporal distribution of animal rabies in Thailand, 2017–2018.** (A) the histogram of rabies in 2017 by month, (B) the spatial distribution of rabies in 2017, (C) the spatial clusters of rabies in 2017, (D) the histogram of rabies in 2018 by month, (E) the spatial distribution of rabies in 2018, and (F) the spatial clusters of rabies in 2018. Our base map is retrieved from the Land Development Department, Thailand (http://www.lddservice.org/lddapp/client/#/map).

occurred in roughly the same areas but covered a wider range of distribution (Fig 1E). Our cluster analysis revealed 19 clusters of animal rabies cases in 2017 (Fig 1C) and 20 clusters in 2018 (Fig 1F). The details of cluster analysis were shown in Tables A and B in S1 Tables.

## Spatio-temporal modeling

Parameters and outputs from the 100 bootstrapped models were averaged (Table 2). Nine statistically significant variables (p-value < 0.05) were identified namely, rabies cases found within the sub-district in the past month, rabies cases found in neighboring sub-district(s) in

**Table 2. Average outputs of the GAMs for the spatial factors for animal rabies occurrences in Thailand, 2017–2018 (100 bootstraps).**

| Spatial factors | Mean | Std. error | P-value | lci* | uci** |
|---|---|---|---|---|---|
| Intercept | -2.77 | 0.054 | < 0.0001 | -2.77 | -2.78 |
| Rabies occurred in the sub-district in the past month | 2.31 | 0.191 | < 0.0001 | 2.27 | 2.34 |
| Rabies occurred in neighboring sub-district (s) in the past month | 1.68 | 0.110 | < 0.0001 | 1.66 | 1.69 |
| Length of the main road | 2.89 | 3.332 | < 0.0001 | 2.43 | 3.35 |
| Length of the secondary road | 2.18 | 2.725 | 0.0155 | 2.07 | 2.28 |
| Distance to the country border | 8.80 | 8.986 | < 0.0001 | 8.79 | 8.80 |
| Density of the owned dog population | 5.71 | 6.777 | 0.0105 | 5.54 | 5.89 |
| Density of human population | 5.82 | 7.006 | < 0.0001 | 5.71 | 5.93 |
| Density of cattle population | 4.62 | 5.709 | < 0.0001 | 4.54 | 4.70 |
| The month of rabies occurrence | 6.46 | 7.464 | < 0.0001 | 6.21 | 6.72 |

*lci = lower confidence interval

**uci = upper confidence interval

the past month, the length of the main road, the length of the secondary road, the distance to the country border, the density of owned dog population, the density of human population, the density of cattle population, and the month of rabies occurrence. Besides, we found that 'rabies occurring in neighboring sub-district(s) in the past month' caused the largest impact (Table 3). The association between the fitted functions and the predictor variables is depicted in Fig 2. It appeared that four variables, including rabies occurred within the sub-district in the past month, rabies occurred in neighboring sub-district(s) in the past month, the length of the main road, and the length of the secondary road showed a similar positive association with the predicted values (Fig 2A–2D). In contrast, a negative association was found for the factors of the distance to the border (Fig 2E). The three variables, including the density of owned dog population, the density of human population, and the density of cattle population, showed a positive association with the fitted function at the beginning of the modeled values (with 50 heads / km$^2$ of the density of owned dog population, 4,000 persons / km$^2$ of the density of human population, and 40 heads / km$^2$ of the density of cattle population), and then the association turned negative when the values were higher. The month of rabies occurrence, which is a

**Table 3. Impact of predictors used in the models.**

| Predictors | AIC | R$^2$ | Deviance explained |
|---|---|---|---|
| 1+2+s(3)+s(4)+s(5)+s(6)+s(7)+s(8)+s(9) | 4482.956 | 0.137 | 16.30% |
| 2+s(3)+s(4)+s(5)+s(6)+s(7)+s(8)+s(9) | 4570.276 | 0.116 | 14.80% |
| 1+s(3)+s(4)+s(5)+s(6)+s(7)+s(8)+s(9) | 4642.300 | 0.112 | 13.40% |
| 1+2+s(4)+s(5)+s(6)+s(7)+s(8)+s(9) | 4482.956 | 0.137 | 16.30% |
| 1+2+s(3)+s(5)+s(6)+s(7)+s(8)+s(9) | 4439.164 | 0.148 | 17.10% |
| 1+2+s(3)+s(4)+s(6)+s(7)+s(8)+s(9) | 4521.948 | 0.139 | 15.20% |
| 1+2+s(3)+s(4)+s(5)+s(7)+s(8)+s(9) | 4451.070 | 0.147 | 16.90% |
| 1+2+s(3)+s(4)+s(5)+s(6)+s(8)+s(9) | 4466.782 | 0.145 | 16.40% |
| 1+2+s(3)+s(4)+s(5)+s(6)+s(7)+s(9) | 4467.222 | 0.146 | 16.70% |
| 1+2+s(3)+s(4)+s(5)+s(6)+s(7)+s(8) | 4480.492 | 0.143 | 16.30% |

1 = RB before, 2 = RB in neighbor before, 3 = Main road, 4 = Secondary road, 5 = Distance to border, 6 = Owned dog density, 7 = Human population density, 8 = Cattle density, 9 = Month, s = smooth function, AIC = The Akaike information criterion

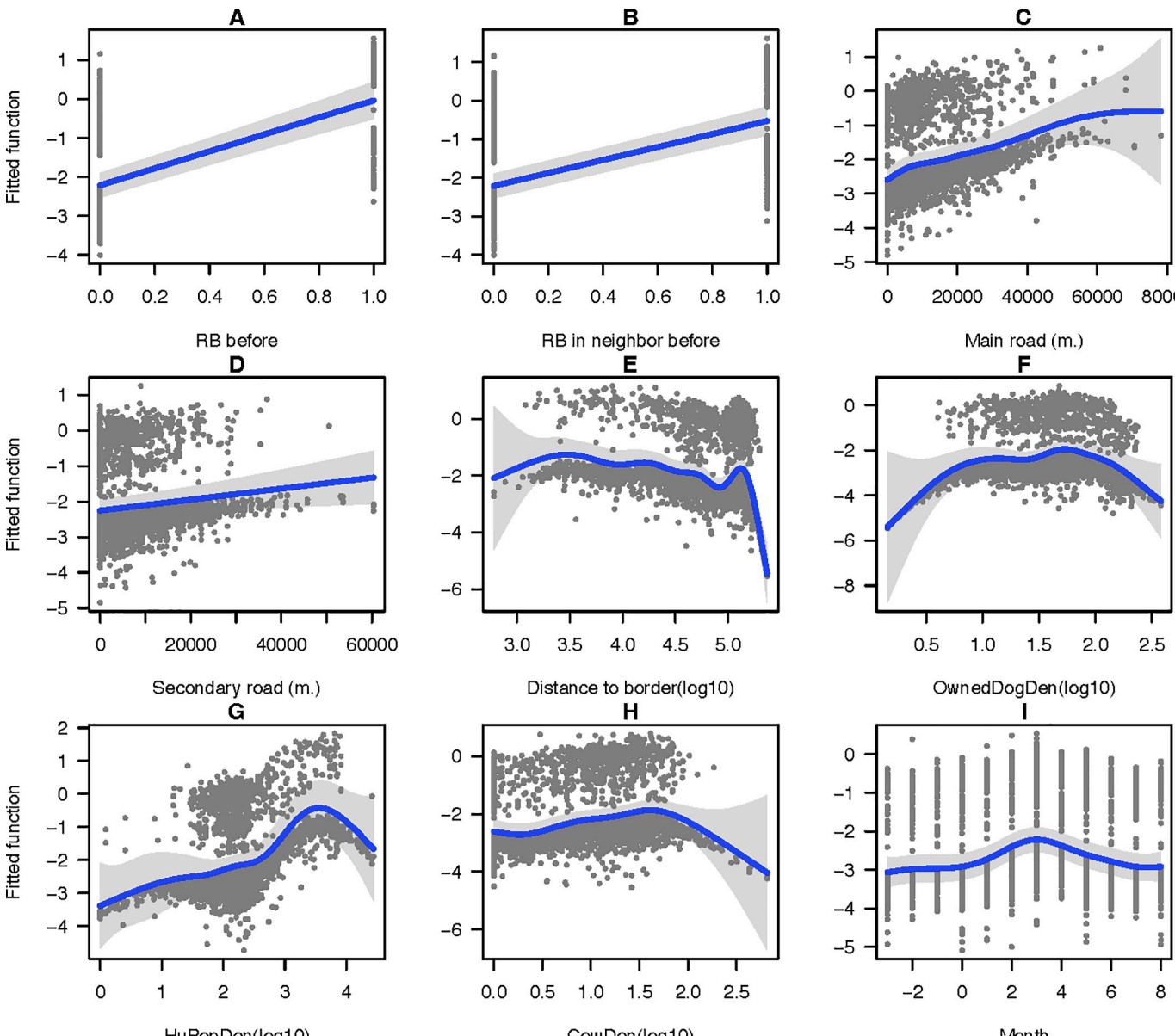

**Fig 2. Fitted function plots of the spatial factors used in the prediction of animal rabies distribution in Thailand.** The fitted function plots of 9 variables including (A) rabies occurred in owned sub-district in the past month, (B) rabies occurred in the neighboring sub-district (s) in the past month, (C) the length of the main road, (D) the length of the secondary road, (E) the distance to the border, (F) the density of owned dog population, (G) the density of human population, (H) the density of cattle population, and (I) the month of rabies occurrence.

temporal variable, showed a positive association between July and January, and, again, the association turned negative between February and June.

The predicted values showed fairly high accuracy for the whole period of model and test sets (Fig 3A) while high variation between low and high accuracy was found in the monthly models. In the whole period, the AUC values of the model set indicated a higher accuracy with a mean of 0.791 (range: 0.784–0.797) (Fig 4A) compared to that of the test sets (mean AUC value = 0.753; range: 0.745–0.759) (Fig 4B). In the validation of the monthly models, the mean of AUC values in the model sets ranged between 0.738 and 0.830 while that of the test sets

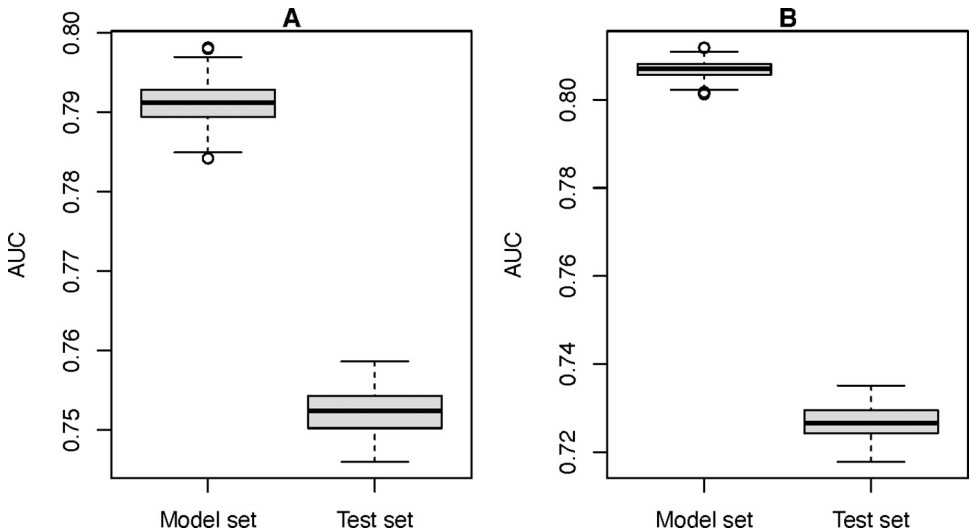

**Fig 3. The predictive power of the model and test sets in the whole modeling period.** A. Data from Feb 2017 –Jan 2018 were used for the model set and data from Feb 2018 –Jan 2019 were used for the test set. B. Vice versa.

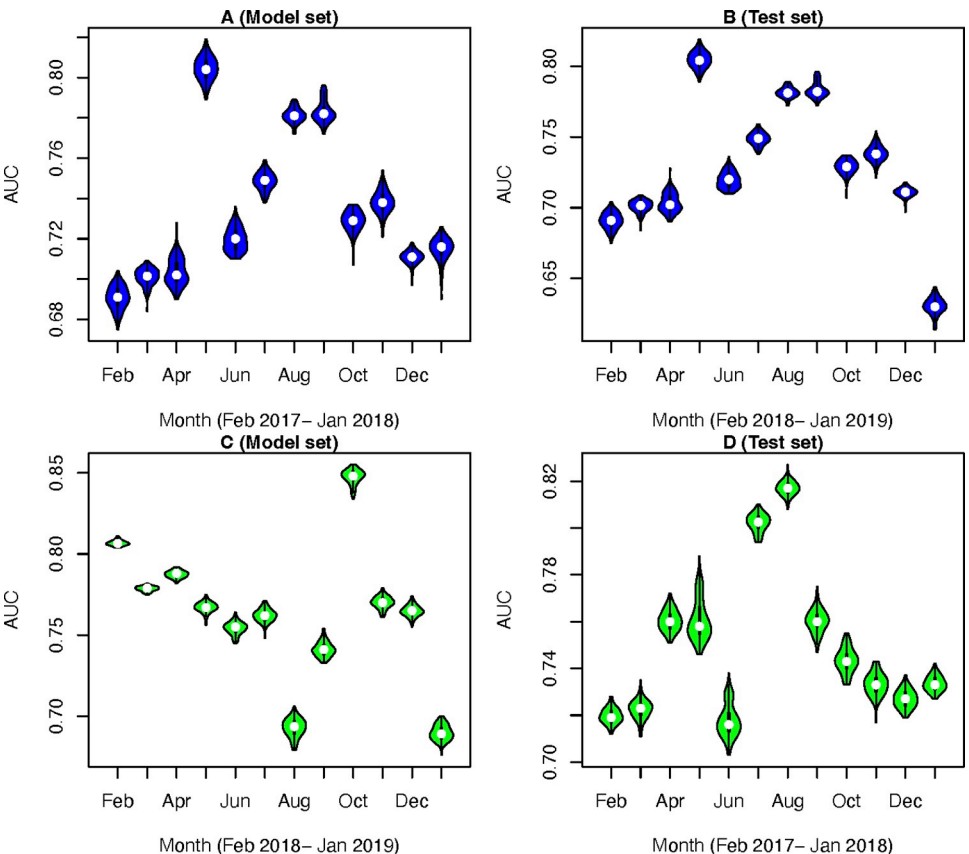

**Fig 4. The predictive power of the monthly model and test sets (100 bootstraps).** A. Model set using data from Feb 2017 –Jan 2018, B. Test set using data from Feb 2018 –Jan 2019, C. Model set using data from Feb 2018 –Jan 2019, and D. Test set using data from Feb 2017 –Jan 2018.

ranged between 0.621 and 0.803. The predicted maps of rabies by months from February 2018 to January 2019 (test sets) are demonstrated in Fig 5. The predicted values (ranged from 0–1) were classified into three classes including; low if predicted value ≤ 0.3, moderate if predicted value > 0.3 and ≤ 0.6, and high if predicted value >0.6. We found that the predicted values of rabies cases in January 2019 were the highest (mean AUC = 0.803; range: 0.798–0.808) with the animal rabies cases highly distributed in the northeast, the central, and the south of the country. In contrast, the lowest accuracy of the predictive values (mean AUC = 0.62; range: 0.600–0.637) was observed in August 2018.

After swapping the model and the test sets, we found that, in the whole period, the AUC values of the model set indicated a higher accuracy with a mean of 0.807 (range: 0.801–0.812) compared to that of the test sets (mean AUC value = 0.727; range: 0.718–0.735) (Fig 3B). In the validation of the monthly models, the mean of AUC values in the model sets ranged between 0.690 and 0.848 (Fig 4C) while that of the test sets ranged between 0.718 and 0.817 (Fig 4D). The results were not much different from the original tests.

## Discussions

This study descriptively characterized the animal rabies outbreaks in Thailand between 2017 and 2018, and further quantified the association between the predictive factors and rabies occurrences at the sub-district level. The outputs were then used to predict the probability of rabies occurrences for one month in advance. Nine spatio-temporal factors influencing the spread of rabies virus in animals in Thailand were identified including eight spatial and one temporal factors (month of rabies occurrence).

We found that the dog was still major rabies-affected species in Thailand and it serves as the main reservoir host maintaining the circulation of rabies virus in the country. This corresponds to previous reports in Thailand [3] and elsewhere [7,20–21,31–32]. Interestingly, most of the rabies-infected dogs were owned, free-roaming or semi-free roaming, and unvaccinated. Besides, the owned dog density was one of the spatial risk factors influencing rabies spread in the country. Our findings were in line with an epidemiological investigation report of rabies outbreaks in Roi-et province, Thailand. The outbreak investigation suggested that most of the infected animals were owned dogs (70%) and roamed freely (76%) [33]. A study in Bali, Indonesia reported that free-roaming owned dogs were 2–3 times more likely unvaccinated compared to those confined (combined ORs: 1.9–3.6, 95% CI: 1.4–5.4) [34]. Free-roaming behavior increases contact rates among dogs [35,36] and hence results in a higher chance of rabies exposure. Consequently, the risk of rabies infection was inevitably increased, especially among those unvaccinated [37–39]. Moreover, a high turnover rate was observed among free-roaming dogs living in high dog density areas causing difficulty in dog population management and rabies control [40]. Therefore, the best way to solve the problem is to reduce the number of free-roaming dogs by promoting the activities on dog ownership, such as reducing the reproductive capacity of the owned dog population, enforcement of mandatory dog identification, and promoting dog-keeping practices [8,39,41].

Several predictive factors positively associated with rabies occurrences reflected urban areas such as length of the main and secondary roads and density of dog and human population. A study on how environmental features or global transport networks influence the pathogen invasion and spread in Tanzania addressed the role of the roads as a facilitator for viral spread [42]. This may be a highly potential factor for the spread of the rabies virus in companion animals, particularly in cities where transport networks are highly connected. Concerning density, the dog population density is a common feature associated with the transmission rate of rabies as suggested in different modeling studies [43–46]. Likewise, the human population

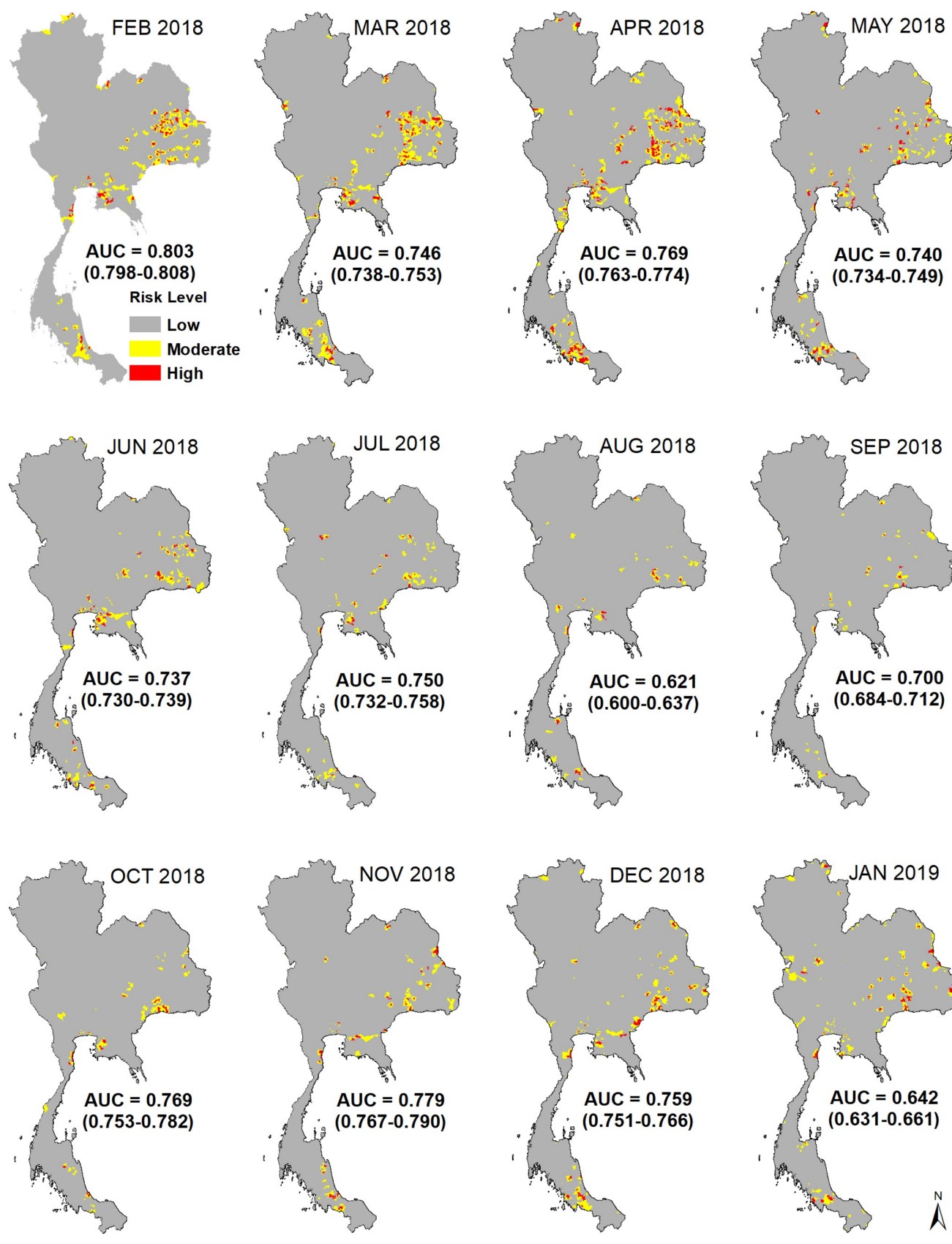

**Fig 5. The predictive maps of rabies risk by months in Feb 2018 –Jan 2019.** The risk was classified into 3 levels including low (predicted value ≤ 0.3), moderate (predicted value > 0.3 and ≤ 0.6), and high (predicted value > 0.6). Our base map is retrieved from the Land Development Department, Thailand (http://www.lddservice.org/lddapp/client/#/map).

density is generally considered a common factor associated with the spatial distribution of many infectious diseases [14–19]. Moreover, a high density of the dog population may result in a high turnover rate, leading to the reduction of vaccine coverage in the areas [40].

Effective control practices are necessary for the outbreak areas. However, different rabies prevention and control strategies have been consecutively implemented in Thailand, for instance, reactive ring vaccination, public relation, elimination of unvaccinated animals possibly exposed to the infected cases [5]. The recent occurrence of animal rabies cases within and among neighboring sub-districts was identified as a risk factor in this study. Therefore, more efforts should be made to contain the spread of the virus once notified. The size of ring vaccination surrounding the cases should be adjusted according to the duration of onset and density of the hosts. Additionally, vaccination campaigns should be robustly implemented [37]. Besides, post-outbreak surveillance programs should be strengthened, for example, continuous monitoring of other animals that are possibly exposed to the cases and stabilizing the number of animals in the outbreak areas especially free-roaming dogs [33].

Based on our findings, the monthly predictive maps allow us to identify the high-risk areas where higher intensive interventions are required. Regular pulse vaccinations and intensively stabilizing the number of free-roaming dogs are highly recommended in these areas. An annual vaccination campaign is a common strategy carried out in Thailand and other rabies endemic countries across the globe. However, the pulse vaccination strategy, in which animals identified in the risk group are repeatedly vaccinated, should be introduced into the endemic areas. A previous study suggested that regular pulse vaccinations would be implemented to maintain a sufficient immunity level against rabies virus in the populations particularly in the areas with high turnover rates [47]. The targeted populations would be firstly prioritized to the owned dogs that are easier to be restrained compared to the ownerless ones [43]. This possibly reaches the WHO vaccination target of 70%, in which the herd immunity is enough to control or eliminate rabies in the areas [43]. This study also suggests that rabies outbreaks reached the peak at the beginning of the year. Therefore, the end of the year would be an appropriate time for implementing an additional vaccination program. The livestock losses remain high in Asia [48] and Thailand as found in this study. Rabies vaccination in livestock particularly cattle in the endemic areas should be considered [48–50]. Stabilizing the number of free-roaming dogs, particularly in the endemic areas, should be strongly implemented, focusing on young, female dogs. A previous individual-based model strongly suggested that neutralizing only young, female dogs was the best population control strategy (reducing 90–91%) compared to targeting female dogs of any age or mixed-sex sterilization of only young dogs (reducing 82–92%) [51].

The main limitation of this study was the incompleteness and unavailability of some data. As appeared in the descriptive analysis of rabies cases derived from Thairabies.net, several important variables were unidentified such as ownership (12%), vaccination history (50.4%), and animal-keeping practices (69.1%). In our spatial modeling, some important data, such as vaccine coverage, were unavailable and/or incomplete. This is one of the plausible explanations for why low accuracy was found in some of our predictive maps. The improvement of robust routine data collection is required for better analysis and prediction in future studies. Moreover, we focused on the changes that occurred during the epidemic years and we tried to produce a tool for a short time prediction based on spatial characteristics of a certain area. Therefore, our model did not reflect the long-term space-time patterns and seasonality. A

future study working on more longitudinal data is recommended. Nevertheless, the data we used here was based on the epidemic years in which the surveillance activities may increase due to a higher number of diagnoses and reported cases. The data on non-epidemic years should be also considered in the future study.

What we observed from our study would be useful to guide relevant public health policies. For example, in public education, we should ask people to pay more attention to the owned dogs rather than only strays. Besides, annual canine rabies vaccination should be rigorously emphasized among dog owners. In terms of disease control, veterinary authorities should focus more on repeated outbreak areas with densely human populations as the risk of recurrence is high. Ultimately, our modeling framework is readily integrable into a real-time surveillance system to produce a timely predictive model resulting in a better preparedness intervention.

## Conclusions

This study described the characteristics of animal rabies cases, in which most of the affected animals were owed, free or semi-free roaming, and unvaccinated dogs. We exploited GAMs to predict the occurrences of rabies at a sub-district level for one month in advance. Our modeling technique is integrable with the dynamic database collecting systems. The dynamic and automatic predictive systems of rabies transmission can be then established. Our predictive maps are applicable for rabies prevention and control as well as strengthening the surveillance system in Thailand and other endemic countries with similar settings. Moreover, the identified risk factors should be rigorously considered and integrated into the strategic plans for the prevention and control of animal rabies in Thailand.

## Supporting information

**S1 Tables.** Table A in S1 Tables Cluster analysis of animal rabies cases in Thailand in 2017. Table B in S1 Tables Cluster analysis of animal rabies cases in Thailand in 2018. (PDF)

## Acknowledgments

The authors thank the Department of Livestock Development, Thailand for providing all necessary data used in our analysis.

## Author Contributions

**Conceptualization:** Weerapong Thanapongtharm, Marius Gilbert, Anuwat Wiratsudakul.

**Data curation:** Weerapong Thanapongtharm.

**Formal analysis:** Weerapong Thanapongtharm.

**Funding acquisition:** Anuwat Wiratsudakul.

**Investigation:** Weerapong Thanapongtharm, Arun Chumkaeo, Anuwat Wiratsudakul.

**Methodology:** Weerapong Thanapongtharm, Marius Gilbert.

**Project administration:** Sarin Suwanpakdee, Anuwat Wiratsudakul.

**Resources:** Weerapong Thanapongtharm, Arun Chumkaeo.

**Software:** Weerapong Thanapongtharm.

**Supervision:** Marius Gilbert, Anuwat Wiratsudakul.

**Validation:** Weerapong Thanapongtharm, Anuwat Wiratsudakul.

**Visualization:** Weerapong Thanapongtharm.

**Writing – original draft:** Weerapong Thanapongtharm, Anuwat Wiratsudakul.

**Writing – review & editing:** Weerapong Thanapongtharm, Sarin Suwanpakdee, Anuwat Wiratsudakul.

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
