## [Decision Letter · Decision Letter 0]

10 Jun 2021

Dear Dr. Wiratsudakul,

Thank you very much for submitting your manuscript "Current characteristics of animal rabies cases in Thailand and relevant risk factors identified by a spatial modeling approach" for consideration at PLOS Neglected Tropical Diseases. As with all papers reviewed by the journal, your manuscript was reviewed by members of the editorial board and by several independent reviewers. In light of the reviews (below this email), we would like to invite the resubmission of a significantly-revised version that takes into account the reviewers' comments. 

We cannot make any decision about publication until we have seen the revised manuscript and your response to the reviewers' comments. Your revised manuscript is also likely to be sent to reviewers for further evaluation.

Sincerely,

Marilia Sá Carvalho

Associate Editor

Guilherme Werneck

Deputy Editor

Reviewer's Responses to Questions

**Key Review Criteria Required for Acceptance?**

**Methods**

-Are the objectives of the study clearly articulated with a clear testable hypothesis stated?

-Is the study design appropriate to address the stated objectives?

-Is the population clearly described and appropriate for the hypothesis being tested?

-Is the sample size sufficient to ensure adequate power to address the hypothesis being tested?

-Were correct statistical analysis used to support conclusions?

-Are there concerns about ethical or regulatory requirements being met?

Reviewer #1: - at line 121 authors provide the general formula for a GAM model. However, it’s note possible to check which was the final model used to produce the results. Based on the figure 2 seems that the smooth function was used for Main road, Distance to border, Density of Owned Dog, Human Population Density, Density of cattle population and Month of Rabies occurrence. Is it correct? which function did the authors used for each of these?

- Authors uses the term Spatiotemporal. However, while is possible to see that author using a kind of connectivity matrix, which I believe that was used to lead with spatial auto-correlation the same is not true for time. Could the authors explain better how did address the problem. Again, I believe that specifying the final model could be very helpful here

Reviewer #2: The objectives are clearly articulated with the hypothesis presented.

The design of the study is partially appropriate.The analysis of spatio-temporal patterns were compromised due to the short period of time worked by the authors

The population is well described and appropriate for the hypotheses being tested, despite the limitations inherent in the use of secondary data from surveillance systems.

Due to the time interval used (2 years), the sample size is insufficient to guarantee adequate power to address the tested hypotheses. In addition, the authors did not present a sample design plan.

The statistical analysis used is adequate to support the conclusions.

Not describe the necessary ethical authorizations for obtaining animal samples.

**Results**

-Does the analysis presented match the analysis plan?

-Are the results clearly and completely presented?

-Are the figures (Tables, Images) of sufficient quality for clarity?

Reviewer #1: - I believe that table two will be more informative if the authors present point estimates and confidence interval instead of p-value.

- Generally when you try to set a predictive model is common to check which features have the biggest impact on predictions. Could you please provide this information.

Reviewer #2: The analyzes presented are in accordance with the analysis plan, despite the limitations indicated in the study methodology (Short time interval and use of secondary bases from local surveillance systems)

The results are presented in a clear and complete manner, despite the lack of information in the initial descriptive analysis, for some variables obtained from the surveillance bases

The figures shown are of sufficient quality.

**Conclusions**

-Are the conclusions supported by the data presented?

-Are the limitations of analysis clearly described?

-Do the authors discuss how these data can be helpful to advance our understanding of the topic under study?

-Is public health relevance addressed?

Reviewer #1: (No Response)

Reviewer #2: The methodological limitations of the manuscript directly impact the quality of the results presented. The authors did not clearly present the limitations of the analysis, such as the incompleteness of the data obtained in the surveillance services, and the short time used in the analyzes, which substantially prevents adequate interpretations of space-time patterns and seasonality.

In addition, the authors do not adequately discuss how the data can be useful to advance the understanding of the topic under study, limiting themselves to bringing local and regional aspects; And public health relevance has been poorly addressed, and could be further developed.

**Editorial and Data Presentation Modifications?**

Reviewer #1: - change model set to train set

Reviewer #2: Include information on ethical procedures related to obtaining animal samples (if available).

To address in the discussion global aspects related to the findings found in the study and bring the relevance of the conclusions to the field of public health.

**Summary and General Comments**

Reviewer #1: In the study “Current characteristics of animal rabies cases in Thailand and relevant risk factors identified by a spatial modeling approach” Wiratsudakul et al. describe the application of GAM model for generate risk maps of rabies using secondary data from Thailand. The article is well written and built on a sound premise and hypothesis. The results are interesting and can be relevant, specially for public health and those interested in applying statistical/computational methods to improve health surveillance. However there are some major points that need to be clarified before the same could be published

Reviewer #2: Original study with relevance for the area, and of interest to researchers, professionals and policy makers specific to animal and human rabies. Apparently, there is no potential interest for researchers or professionals outside the area. Partially rigorous methodology, with conclusions justified by the evidence presented, but with important limitations in the methodology that directly impact the interpretation of the results presented.

There was no mention in the manuscript of ethical approvals in the animal field.

It would be very important extend the study period for the conclusions about temporal space patterns become robust. Working with only two years, limits interpretation of the findings, since only "epidemic" years were included. This is also reflected in the active / passive surveillance service, which, theoretically, may be more attentive and agile in recent years, due to the significant increase in diagnosed / reported cases.

In the introduction, the authors provide a historical overview of notifications of animal and human rabies in the studied territory. It is suggested to include in the analysis a broader period of time, for example, 10 years, so that the historical series includes a significant period of time, which better reflects the reality of rabies in the territory.

PLOS authors have the option to publish the peer review history of their article (what does this mean?). If published, this will include your full peer review and any attached files.

Reviewer #1: No

Reviewer #2: No
---

## [Decision Letter · Decision Letter 1]

5 Nov 2021

Dear Dr. Wiratsudakul,

We are pleased to inform you that your manuscript 'Current characteristics of animal rabies cases in Thailand and relevant risk factors identified by a spatial modeling approach' has been provisionally accepted for publication in PLOS Neglected Tropical Diseases.

Best regards,

Marilia Sá Carvalho

Associate Editor

Guilherme Werneck

Deputy Editor

Reviewer's Responses to Questions

**Key Review Criteria Required for Acceptance?**

**Methods**

-Are the objectives of the study clearly articulated with a clear testable hypothesis stated?

-Is the study design appropriate to address the stated objectives?

-Is the population clearly described and appropriate for the hypothesis being tested?

-Is the sample size sufficient to ensure adequate power to address the hypothesis being tested?

-Were correct statistical analysis used to support conclusions?

-Are there concerns about ethical or regulatory requirements being met?

Reviewer #1: The objectives are clearly described with the statistical analysis adequate to answer the hypothesis presented.

Reviewer #2: The objectives are clearly articulated with the hypothesis presented.

The study design is appropriate, as it aims to meet the proposal of analyzing monthly spatiotemporal patterns.

The population is well described and appropriate for the hypotheses being tested, despite the limitations inherent in the use of secondary data from surveillance systems.

The sample size is sufficient to ensure adequate power to address the tested hypotheses.

The statistical analysis used is adequate to support the conclusions.

**Results**

-Does the analysis presented match the analysis plan?

-Are the results clearly and completely presented?

-Are the figures (Tables, Images) of sufficient quality for clarity?

Reviewer #1: The analysis used in the paper is adequate with the objectives pruposed.

Reviewer #2: The analyzes presented are in accordance with the analysis plan.

The results are presented clearly and completely.

The figures shown are of sufficient quality.

**Conclusions**

-Are the conclusions supported by the data presented?

-Are the limitations of analysis clearly described?

-Do the authors discuss how these data can be helpful to advance our understanding of the topic under study?

-Is public health relevance addressed?

Reviewer #1: The conclusion are supported by the results found

Reviewer #2: The methodological design presented is of high quality and offers adequate results.

The analyzes performed are suitable for interpretations of spatiotemporal patterns and seasonality.

Authors adequately discuss how data can be useful to advance the understanding of the topic under study, bringing local and regional aspects of relevance; Public health impact was adequately addressed.

**Editorial and Data Presentation Modifications?**

Reviewer #1: (No Response)

Reviewer #2: (No Response)

**Summary and General Comments**

Reviewer #1: (No Response)

Reviewer #2: Original study with relevance to the area, and of interest to researchers, practitioners and policy makers specific to animal and human rabies. There is potential interest for researchers or professionals from outside the area. Rigorous methodology, with conclusions justified by the evidence presented. Researchers adequately justified the methodological proposal presented. And the results were properly interpreted.

Researchers justified the absence of ethical approvals in the animal area.

PLOS authors have the option to publish the peer review history of their article (what does this mean?). If published, this will include your full peer review and any attached files.

Reviewer #1: No

Reviewer #2: No

---

## [Editor Report · Acceptance letter]

22 Nov 2021

Dear Dr. Wiratsudakul,

We are delighted to inform you that your manuscript, "Current characteristics of animal rabies cases in Thailand and relevant risk factors identified by a spatial modeling approach," has been formally accepted for publication in PLOS Neglected Tropical Diseases.

Best regards,

Shaden Kamhawi

co-Editor-in-Chief

Paul Brindley

co-Editor-in-Chief
